# Effects of a Digitally-Enabled Healthy Eating and Physical Activity Diabetes Prevention Peer Support Program on Weight over 6-Months

**DOI:** 10.3390/nu17223599

**Published:** 2025-11-18

**Authors:** Freya MacMillan, Holly Hliounakis, Kayla Jaye, Kimberly Mitlehner, Chris Pitt, Kate A. McBride, Uchechukwu Levi Osuagwu, David Simmons

**Affiliations:** 1Translational Health Research Institute, Western Sydney University, Sydney, NSW 2751, Australia; 2Diabetes Obesity Metabolism Translational Research Unit, Western Sydney University, Sydney, NSW 2751, Australia; 3Division of Research and Innovation, Western Sydney University, Sydney, NSW 2751, Australia; 4Macarthur Clinical School, School of Medicine, Western Sydney University, Sydney, NSW 2751, Australia; 5Charles Perkins Centre, University of Sydney, Sydney, NSW 2006, Australia; 6Bathurst Rural Clinical School, Western Sydney University, Bathurst, NSW 2795, Australia

**Keywords:** diabetes mellitus, type 2, health behavior, peer group, digital health, digital technology, weight loss, secondary prevention

## Abstract

Background/Objectives: Type 2 diabetes (T2D) is a growing health epidemic. Innovative approaches such as digital technologies incorporating peer-supported coaching have shown promise in diabetes prevention. This study aimed to examine the feasibility and effect on weight of a digitally-enabled peer support program in inner-regional Sydney. Methods: A pre-post study of a digitally-enabled peer support initiative promoted weight management and lifestyle changes in participants at risk of T2D in inner-regional Sydney. Participants were recruited primarily from general practices and community groups. Participants received initial guidance, educational videos, goal-setting tools, and self-assessment weights, while volunteer peer support facilitators provided ongoing support through action planning and monthly calls. Baseline and follow-up weights at 6 months were collected to determine program effectiveness, while feasibility was evaluated through short exit interviews and analytic website data. Results: Most eligible participants (92.4%) were recruited through general practice. Program completers (*n* = 35, 43.8%) reported an average weight reduction of 3.7 kg (SD = 3.9, *p* < 0.001). Those who used the platform to log at least one achievement saw a greater reduction in weight than those who did not log achievements (mean difference = −2.9 kg, 95% CI −5.6 to −0.1, *p* = 0.049). Exploratory qualitative analysis of exit interviews revealed challenges surrounding technology, website interaction, scheduling conflicts, data collection, and attrition. Conclusions: Preliminary results indicate that this digital program was associated with significant weight reduction among individuals at risk of diabetes in an inner-regional area of Sydney. Recruitment was most effective via general practices, highlighting the potential for such a program to be promoted through this setting.

## 1. Introduction

Type 2 diabetes (T2D) has been estimated to account for greater than 90% of all diabetes diagnoses world-wide [1], and affects about 8.8% of the global population [2]. In Australia, the total diagnoses are estimated to reach 2.0–2.9 million by the end of 2025 [3]. The annual economic burden of T2D in Australia has been estimated at AUD $14.6 billion, with individual direct costs greater in those experiencing diabetes-related micro-and macrovascular complications [4], such as nephropathy and cardiovascular disease. Overweight and obesity are key risk factors contributing to this burden [5], however, meeting the national dietary [6] and physical activity recommendations [7] can significantly reduce these, mitigating the risk of developing T2D.

Previous studies have shown that diabetes prevention programs are able to reduce the overall risk of T2D [8], with up to a 58% risk reduction among high-risk participants [9]. In particular, studies have reported weight loss as a dominant predictor of reductions in incidence of T2D and associated cardiometabolic risk factors including glucose and HbA1c [10,11]. Additionally, the effect of structured lifestyle diabetes prevention interventions can last up to 20 years post-receiving the active intervention [12], further supporting the implementation of lifestyle modification strategies. Whilst such lifestyle change programs are effective, these programs tend to recruit those most motivated to participate in trials and often fail to recruit harder to reach groups.

Face-to-face peer support delivered by peer support facilitators (PSFs) also shows great promise for improving lifestyle behaviours and preventing diabetes [13,14]. A community-based, church peer support program in south-western Sydney (SWS) reported statistically significant reduction in HbA1c over 3–8 months in the total church sample, as well as in those with existing diabetes [13]. Additionally, weekly total time spent in moderate and vigorous physical activity levels and diabetes knowledge increased post-intervention [13]. This reflects broader findings from a systemic review of available peer support programs, which noted a significant reduction in HbA1c levels in six of the nine reviewed studies [14]. Additional findings included improved blood pressure and glycemia in individuals with T2D [14], thus reducing their risk of developing complications of diabetes. The use of peer strategies is therefore a promising approach for reducing the risk of developing T2D in at-risk individuals.

Although previous interventions have been effective in reducing T2D risk factors and overall risk, access and reach to non-urban communities has been limited. These communities are disproportionally affected in Australia, with overall prevalence of T2D approximately two times higher in ‘Remote and Very Remote’ areas in comparison to ‘Major Cities’ [5]. Additionally, they often face lower quality and affordability of healthy foods, as well as limited access to and availability of physical activity opportunities, which contributes to increased T2D risk [15,16,17]. Prior research within inner-regional SWS has identified long commutes to in-person peer meetings as an additional barrier to diabetes prevention program attendance [18]. This highlights a need for alternative ways to engage this population.

Interventions centred around digital technologies could potentially address the associated challenges with reaching geographically dispersed individuals, while offering platforms for scaling peer support. A meta-analysis of digital interventions for the management of T2D reported a greater reduction in HbA1c associated with digital interventions in comparison to standard care, with these studies finding that digital interventions were effective, feasible, and acceptable to participants in relation to this outcome [19]. This study also found greater improvements in studies including higher intensity coaching support, in comparison to those with lower intensity [19]. A longitudinal study that sampled 26,743 participants observed a greater proportion of in-person participants achieving >5% of weight loss, compared to online participants, however, weight-loss retention was significantly greater for online participants [20]. Despite some inconsistency regarding these outcomes, digital interventions have shown promise in T2D prevention and may be more effective when supported through coaching or peer reinforcement strategies.

To date, there is limited research that evaluates the effectiveness and feasibility of digital, peer supported interventions in reducing T2D risk in Australia. Furthermore, inner-regional communities, who face distinct barriers to in-person care, have not been a focus of these interventions. To address these gaps, this pre-post study aimed to assess the feasibility, and effect on weight of a digitally enabled, scalable, peer support program in inner-regional Sydney (Wollondilly) and surrounding suburbs in the Wollondilly Diabetes Programme-Lifestyle Plus study. We hypothesized there would be a reduction in weight of participants in this study, contributing to an overall reduced risk of T2D development in these individuals.

## 2. Materials and Methods

### 2.1. Study Design

This study employed a pre-post design, with data collected at baseline and post-program (6 months). The study was approved by the Western Sydney University Human Research Ethics Committee (approval number H11826) and was a sub-study embedded within a wider project evaluating an integrated-care service—the Wollondilly Diabetes Programme (WDP) [21,22]. WDP is a T2D prevention and clinical service that commenced in 2016. In 2023, the service expanded to offer the digital-enabled peer support program, Wollondilly Diabetes Programme-Lifestyle Plus, which was the focus of this paper.

### 2.2. Participants and Recruitment

Participants were recruited from GPs, community groups, and workplaces within Wollondilly and surrounding suburbs of inner-regional Sydney between March 2023 and April 2024. GPs who assisted with recruitment agreed to apply filters to identify potentially eligible participants. These patients were then invited to participate in the study via email. For recruitment through community groups and workplaces, flyers were circulated in physical locations where community members would be present, as well as content being shared on social media and at community events. Individuals were eligible if they were ≥18 years old and had either an Australian Diabetes Risk Assessment (AUSDRISK) score ≥ 10 [23] and/or prior gestational diabetes mellitus, or IGT or IFG. Peer support facilitators were recruited from community events, GPs, a postgraduate Diabetes program, and through local networks.

### 2.3. Peer Support Program

This program involved a digitally enabled peer support program that had previously been delivered in the UK [24,25]. Prior to program delivery in Australia, the platform was adapted for the Australian context by reflecting foods commonly eaten in Australia and use of appropriate spelling and terminology. Platform adaptation was guided by a dietitian.

Participants were presented with a brief overview of the platform when meeting with the research team for the baseline data collection. Once participants were onboarded, they received a participant session guide outline and link via email to set up their account. Once activated, participants were instructed to watch the educational videos and complete their mandatory initial self-assessment. Following this, each participant was paired with a PSF to complete their first action planning call. Over the six-month period, participants had unlimited access to the online lifestyle behaviour change platform, which included individualized behaviour change tools and goal setting related to diet and physical activity (Figure 1). The goal setting component was supported by self-monitoring tools, including weekly achievement logs for weight and steps, and by educational resources and messages.

### 2.4. Peer Support Facilitation

After being recruited, PSFs participated in a two-hour training session delivered by a dietitian and supported by a research team member, which included an overview of PSF-specific skills and a comprehensive platform overview. They were then invited to participate in regular group meetings, and one-on-one support was provided where necessary. Using session guides that outlined the program timeline and structure, PSFs provided concurrent support to program participants through action planning and monthly calls.

### 2.5. Data Collection

An initial questionnaire was distributed via Qualtrics (Qualtrics; Provo, UT, USA) to screen out individuals who had already been diagnosed with diabetes. For the remaining eligible participants, a REDCap (Vanderbilt University, Nashville, TN, USA) survey was distributed, where eligible participants provided written consent prior to data collection regarding demographic characteristics, diabetes risk [23], and readiness to change [26]. Weight measurements were repeated at program end (6 months post-baseline) and were conducted by either a research staff member, a GP, or self-reported by the participant. At the program end, all participants and peer support facilitators who agreed to provide feedback on their perceptions and experiences of the intervention were sent the same set of questions (participant and PSF questions were tailored to the respective audience) via email and were asked to provide their written feedback.

Charder MS 4600 digital scales (Hamburg, Germany) were used by the research team to collect weight (kg). Heavy shoes and clothes were removed prior to measurements being taken. Alternatively, participants could have their weight measured at their general practice or measured their own weight at home, reporting their weight to the research team. Participants also self-reported height (cm). Body Mass Index (BMI) was calculated and standard cut-points for adults used to categorise BMI [27].

Analytic data from the online platform (i.e., physical activity/dietary goals being met) was collected to understand the engagement and utilisation of the website, based on information and goals logged by the participants.

### 2.6. Statistical Analysis

Statistical analyses were conducted using R statistical software (version 4.4.3, R Foundation for Statistical Computing, Vienna, Austria) [28]. Baseline demographic data and platform usage data were aggregated using means, standard deviations, and percentages. Two-sample t-tests and Pearsons’s chi-square tests were conducted to compare demographic differences between participants who completed the program and those who dropped out. Readiness to change was summarised as the percentage of participants who indicated being in the ‘action’ or ‘maintenance’ stage of change. Weight change between program baseline and end was analysed using means, standard deviations, and paired t-tests, and an intention-to-treat analysis using last observations carried forward was conducted to test the robustness of our findings. Weight change by platform usage was compared using pooled t-tests, with results reported as mean differences and 95% confidence intervals. Statistical significance was set at 0.05. Additionally, conceptual content analysis was used to explore the concepts relating to challenges and feasibility of the intervention within email text responses and to quantify these [29].

## 3. Results

### 3.1. Peer Support Facilitators

Twenty volunteer PSFs were originally recruited for the study, however, only 10 PSFs delivered the program as some dropped out (*n* = 3) or did not engage in the program post-training (*n* = 7). The 10 PSFs that delivered the program were recruited through a postgraduate diabetes course (*n* = 6) for non-diabetes specialist health professionals, local GPs (*n* = 2), a local university (*n* = 1), and a community day event (*n* = 1).

### 3.2. Participant Demographics

Eligible participants (*n* = 118) were recruited via GPs (*n* = 109), Men’s Sheds (*n* = 4), self-referral (*n* = 3), Facebook (*n* = 1) and the local council (*n* = 1). At baseline, participants’ weight was measured by research staff (*n* = 72), were self-reported (*n* = 1) or measured by a GP (*n* = 6). Of these eligible participants, 80 (67.8%) onboarded the platform, with 79 (66.9%) of these completing all baseline data collection (Figure 2). Baseline demographics of the study sample are described in Table 1. On average, participants were aged 58.5 ± 12.3 years and weighed 99.1 ± 19.5 kg. Forty-four percent were men, and 5.1% were of Indigenous Australian background.

43.8% of program starters (*n* = 35) completed the program, including post-program (6-month) follow-up data collection on weight. Twenty-seven (77.1%) had their weight measured by research staff and eight (22.9%) self-reported their weight. There were significantly more male than female participants who completed the program compared to those who did not (62.9% male completers, 29.5% male dropouts; *p* = 0.006; Appendix A). Of the 79 participants who onboarded the platform and completed baseline data collection, 44 participants did not commence the program, dropped out, were unable to be contacted at follow-up, or did not report their weight at follow-up (Figure 2).

### 3.3. Readiness to Change

For dietary questions, readiness to change ranged from 39.2% to 74.7%, with participants most likely to have substituted sugary drinks for water or non-sugary drinks, and least likely to be eating five or more servings of vegetables per day (Table 2). Physical activity readiness changed by about 30.8% to 44.9% with resistance training being the least actioned area and reducing sitting time as the most actioned area. There were no significant differences in readiness to change between genders (Appendix A).

### 3.4. Change in Weight

A significant decrease in weight was observed by the 35 participants who completed the program and had a post-program weight collected (*n* = 27) or self-reported (*n* = 8) (Figure 3). Among these participants, the mean weight loss was −3.7 (SD = 3.9) kg (*p* < 0.001). This equates to a 3.9% (SD = 4.0%) reduction in body weight. The intention-to-treat analysis attenuated these results slightly, with a mean difference of −1.6 (SD = 3.2) kg (*p* < 0.001; Appendix A).

### 3.5. Platform Activity

Of the 35 participants who completed the program, 28 (80.0%), 25 (71.4%), and 19 (54.5%) logged weight, lifestyle behaviour achievements, and steps, respectively, at least once during the program. Twenty-nine (82.9%) participants logged data using at least one of the three features, and 18 (51.4%) participants logged data for all three of the features. On average, participants logged their weight 11 (SD = 12, range 0–54) times, and weekly achievements 7 (SD = 10, range 0–35) times. Participant weight loss was significantly greater for those who logged lifestyles behaviour achievements for at least one week of the program (Δ = −2.9, 95% CI = −5.6 to −0.1). While weight loss was greater for those who used the other features, none of the between-group differences were statistically significant (Table 3).

### 3.6. Program Feedback

#### 3.6.1. Challenges Identified by the Participants and PSFs

Seven participants provided feedback via exit interviews post-program. Several challenges were identified by the participants in this program (Table 4). Some participants struggled to engage with the online platform due to a lack of access to a computer. For participants who could consistently access the platform, some reported that the graphs were difficult to interpret by themselves and reported that their action plans did not always align with the achievements on the online platform. Reliable weight data was a concern due to its self-reported nature. In some cases, scheduling conflicts between the participants and PSFs were an issue, and participants reported a lack of encouragement outside of the monthly scheduled calls with their PSF. One PSF provided feedback, citing challenges regarding limited confidence post-training, and maintaining their confidentiality.

All participants (100%) provided recommendations on how to improve the platform, while four out of seven (57%) provided recommendations on how to improve PSF support. Recommendations included, (1) additional guidance and education on goal setting and action planning, (2) reminder messages and further encouragement between scheduled PSF calls, (3) implementing a buddy system for PSFs post-training to support them, (4) ongoing training and support for PSFs, and (5) improvements to the buddy system.

#### 3.6.2. Challenges Identified by the Research Team

A range of technological challenges arose throughout this program. These included password login issues, frequent timing out and logging out. The research team also identified website interaction and design difficulties such as frequently clicking the ‘back’ button to navigate, and difficulty with the ability to edit or undo self-assessment activities. In addition to these, the team found that many participants were not logging in and using the platform frequently, which was also emphasised by the participant attrition rate (44.3% of the baseline sample who commenced the program). The PSFs reported that the platform was admin-intensive to manage, and that they did not always update their patients’ action plans regularly. Lastly, we found that PSFs had a general focus on helping the participants but did not target their weight loss and had difficulties collecting participant weights.

## 4. Discussion

The primary aim of this study was to evaluate the feasibility and effect of a digitally-enabled peer support program on the weight of individuals at high risk of developing diabetes in inner-regional Sydney. The preliminary findings included a statistically significant, average weight loss of 3.7 kg (SD = 3.9), equating to a 3.9% reduction in body weight from baseline. While these results are preliminary, they potentially have clinical importance. A previous prevention program reported that for each kilogram of weight lost, participants had a 16% lower risk of developing diabetes [10]. Additionally, a cohort study of 5145 reported that those who reported a weight reduction between ≥2% to <5% had significantly greater odds of achieving improvements in systolic blood pressure, glucose, HbA1c, and triglycerides [11]. While direct comparisons are limited due to longer follow-up periods in the other studies, the pattern suggests that the modest, but feasible changes observed within our program may confer meaningful risk reduction. This hypothesis warrants further investigation, with the inclusion of longer follow-up time points, and additional markers of diabetes risk, such as blood pressure, HbA1c, and lipids.

Of the participants who completed the follow-up data collection, 82.9% engaged with the online platform through activity logging. We found that engagement through logging was associated with higher weight loss, particularly for participants who utilized the achievement logging function. Similarly, one systematic review found that greater use of web-accessible tools was associated with greater improvements in diabetes outcomes, and another reported that web logins for e-therapies were associated with improved weight loss, food and vegetable consumption, and physical activity outcomes within the respective interventions [30,31]. These findings highlight the need for ongoing engagement strategies throughout the online program.

Participant reach and engagement were measured by recruitment and attrition rate. The importance of leveraging the existing relationships between primary care providers and their patients was highlighted, as we found the most success through recruitment via GPs. This is likely to be attributable to factors such as patients’ trust and confidence in their GP as well as their views of GPs being a credible source for health-related behaviour change advice [32,33,34]. Despite strong recruitment, only two thirds of eligible individuals completed baseline data collection, and less than half of the participants who started the program completed follow-up measurements. We also found higher attrition in women than men. While high attrition is frequently recognized as a challenge in diabetes and online health interventions [35], the wait period between consenting to participate and being onboarded to the platform may have exacerbated attrition in our study. This emphasizes the need to onboard individuals soon after they express interest to maintain the initial interest and motivation.

We identified multiple challenges to the implementation and feasibility of this program through our exploratory qualitative analysis of exit interviews. Technical challenges, a widely recognized issue with web-accessible tools (2), likely impacted retention and engagement. This is consistent with previous studies that have cited frustration with technology as a theme for dropout [36] and factors such as usability, accessibility, and technical support as influencers to the adoption of technology, especially for older adults [37]. Perceiving the program content as irrelevant, and the ease of understanding and use have also been reported to influence engagement in online digital diabetes prevention studies [36,38]. Lastly, scheduling conflicts and limited support outside of monthly PSF calls hindered program engagement. Scheduling issues have reportedly affected attendance at diabetes clinics [35], and timely, available support has been reported to be proportional to the success of digital interventions [19,39]. While peer support was a facilitator to this program, participants recommended greater accessibility and availability of PSFs as potential enablers to ongoing engagement. Given that most PSFs were volunteer healthcare professionals, future studies should address flexibility in scheduling to increase availability and the consistency of support to participants.

There are several strengths and limitations of this study. A key strength of the program included the use of weight loss as the primary outcome, a predictor of reduced diabetes risk. Despite having some challenges, the digital format of program delivery was virtually accessible to anyone with web-connectivity, providing flexibility and convenience for those in inner-regional areas. The digital format also provides the opportunity for this program to be scaled up to reach a larger sample of at-risk populations. The integration of peer-support is also a strength, with additional emotional and practical support provided beyond a digital-only solution. Despite these strengths, the study was limited in the following ways. Firstly, the pre-post design did not include a control group or randomization, therefore, causal relationships regarding this program’s effect on weight loss cannot be established. The lack of a powered sample size, high attrition (56.3%), and reliance on some self-reported weight data at follow-up means that our results may reflect the more positive experiences of this program and potential selection bias, and may therefore limit the generalizability of the findings. Lastly, the 6-month follow-up did not allow for an evaluation of the long-term sustainability or impact of this program.

Strategies for the engagement and retention of participants are essential so that participants can receive the intended improvements for diabetes-related outcomes. Future research should also explore the effectiveness of this program at a larger scale, as a larger sample size and inclusion of a comparison group would strengthen the program’s evaluation. Cultural adaptations to this program could confer reach to other at-risk groups, including Indigenous Australians and multicultural populations. Finally, the measurement of additional clinical endpoints could demonstrate broader clinical benefits relevant to diabetes prevention, alongside weight loss.

## 5. Conclusions

In conclusion, this digitally-enabled program was feasible and acceptable for individuals at risk of diabetes in an inner-regional area of Sydney, with associated weight reductions providing preliminary indications that warrant further investigation. Recruitment via GPs was the most effective method, but improvements are needed to the platform and peer support model to further support participants throughout the course of the program. Future projects scaling up this intervention should focus on increasing the sample size, including a control group, and measuring additional clinical endpoints.

## Figures and Tables

**Figure 1 nutrients-17-03599-f001:**
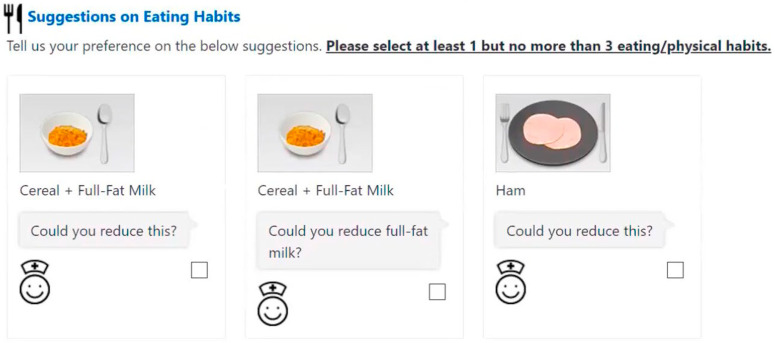
Nutrition-based behaviour change tool from the digitally enabled peer support program.

**Figure 2 nutrients-17-03599-f002:**
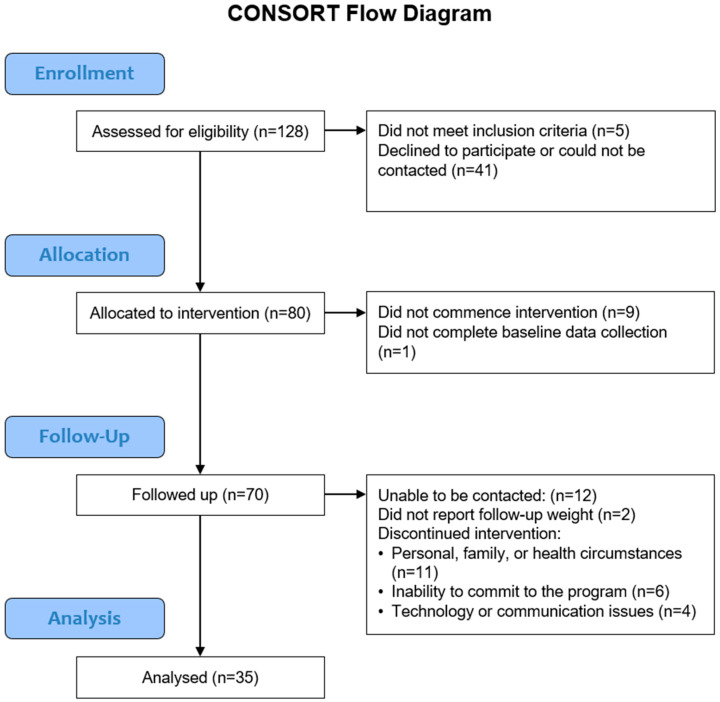
CONSORT diagram.

**Figure 3 nutrients-17-03599-f003:**
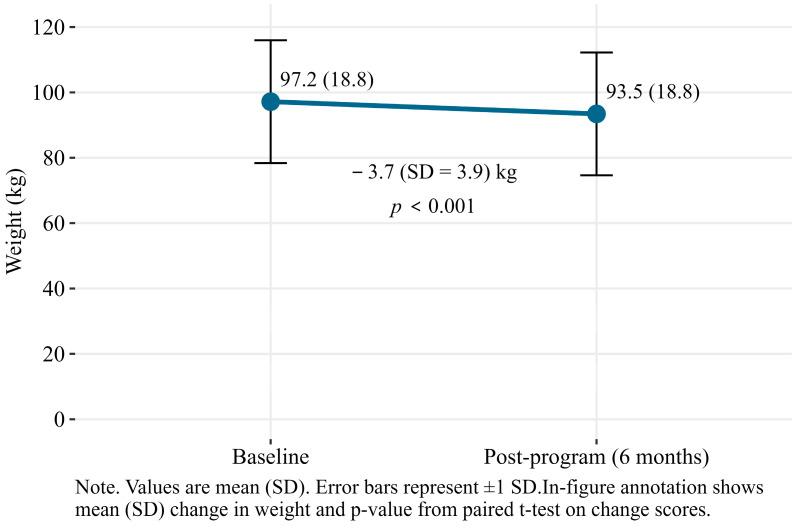
Mean (SD) change in weight; baseline to post-program (6 months).

**Table 1 nutrients-17-03599-t001:** Baseline demographic characteristics.

Characteristic	Responses	Mean (SD) or N (%)
Age	78	58.5 (12.3)
Age group	78	
Under 35 years		4 (5.1%)
35–44 years		4 (5.1%)
45–54 years		21 (26.9%)
55–64 years		25 (32.1%)
65 years and over		24 (30.8%)
Gender (male)	79	35 (44.3%)
Indigenous	79	4 (5.1%)
Height (cm)	68	169.6 (9.2)
Weight (kg)	70	98.6 (19.7)
Waist circumference (cm)	67	109.6 (15.0)
BMI	68	34.4 (6.2)
BMI categories	68	
Normal (18.5–24.9)		2 (2.9%)
Overweight (25.0–29.9)		15 (22.1%)
Obese Class I (30.0–34.9)		20 (29.4%)
Obese Class II (35.0–39.9)		16 (23.5%)
Obese Class III (40 and above)		15 (22.1%)
WHO-5 Wellbeing (1–25)	48	14.3 (5.0)
Alcohol consumption (>2 standard drinks/day)	79	32 (40.5%)
High blood fats	79	46 (58.2%)
Mental health issues	79	31 (39.7%)
Highest level of education	79	
Postgraduate		8 (10.1%)
Graduate diploma or certificate		5 (6.3%)
Bachelor’s degree		14 (17.7%)
Advanced diploma or diploma		18 (22.8%)
Certificate		14 (17.7%)
Secondary school		18 (22.8%)
Other		2 (2.5%)
Occupation	79	
Manager		12 (15.2%)
Professional		15 (19%)
Technician or trade worker		1 (1.3%)
Community or personal service worker		5 (6.3%)
Clerical or administrative worker		10 (12.7%)
Sales worker		3 (3.8%)
Retired		24 (30.4%)
Other		9 (11.4%)

**Table 2 nutrients-17-03599-t002:** Readiness to change lifestyle behaviors at baseline.

Characteristic	N (%)
Dietary behaviour—readiness to change diet	
Do you drink water and other non-sugary drinks instead of sugary drinks/fruit juice?	59 (74.7%)
Do you eat at least five or more servings of vegetables daily?	31 (39.2%)
Do you eat at least two fruits every day?	36 (45.6%)
Do you eat at least two servings of dairy foods every day? *	50 (64.1%)
Do you eat at least three different protein foods every 1–2 days?	37 (46.8%)
Do you eat less fat overall?	41 (51.9%)
Have you reduced amount of food you eat at each sitting?	49 (62.0%)
Do you eat more foods with fibre?	48 (60.8%)
Do you eat less sugary foods and carbohydrates?	46 (58.2%)
Do you eat at regular intervals?	46 (58.2%)
Physical activity—readiness to change physical activity	
Are you making yourself stronger? *	24 (30.8%)
Do you plan more activity in your day?	30 (38.0%)
Do you plan more activity in weekends? **	33 (42.9%)
Have you increased the number of steps you take each day? *	34 (43.6%)
Have you reduced the amount of time you spend sitting? *	35 (44.9%)
Weight—readiness to change weight	
Are you trying to reach your best weight?	54 (68.4%)

Note. N = number of participants in action and maintenance stage. * N = 78 responses. ** N = 77 responses.

**Table 3 nutrients-17-03599-t003:** Participant weight loss by active use of platform goal setting.

	Within-Group Δ, Mean (SD)		Between-Group Δ (95% CI)	*p* Value
	Logged ≥ 1	Did Not Log		
Weight (yes *n* = 28)	−4.0 (4.1)	−2.6 (3.2)	−1.4 (−4.6, 1.7)	0.398
Achievements (yes *n* = 25)	−4.5 (3.9)	−1.7 (3.3)	−2.9 (−5.6, −0.1)	0.049
Steps (yes *n* = 19)	−4.3 (3.7)	−3.1 (4.2)	−1.2 (−3.9, 1.6)	0.376
Weight, achievement, or steps (yes *n* = 29)	−4.1 (4.1)	−1.7 (2.4)	−2.5 (−5.1, 0.2)	0.164
Weight, achievement, and steps (yes *n* = 18)	−4.1 (3.7)	−3.4 (4.2)	−0.7 (−3.4, 2.0)	0.604

Note. Pooled t-tests were used to assess between-group differences. A *p*-value <0.05 was considered statistically significant. Δ = change.

**Table 4 nutrients-17-03599-t004:** Challenges identified by participants, PSFs, and researchers.

Reported Challenge
Participant reported challenges
Low engagement due to lack of access to a computer
Difficulty in interpreting graphs
The action plan did not always align with achievements
Scheduling conflicts with PSFs Lack of encouragement from PSFs outside of the monthly scheduled check-in calls
PSF-reported challenges
Limited confidence post-training Maintaining their confidentiality
Researcher-reported challenges
Technology (password login issues, frequent timing out and logging out)
Website interaction difficulties, particularly with navigation using the ‘back’ button
Difficulty editing or undoing self-assessment activities Infrequent logins and engagement of participants Admin-intensive management of the platform Reaching participants for data collection, specifically for weight
PSFs were generally helpful but did not emphasise participant weight loss enough

## Data Availability

The datasets presented in this article are not readily available because they contain identifiable participant information and are subject to privacy and confidentiality restrictions. Requests to access the datasets should be directed to f.macmillan@westernsydney.edu.au.

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
