# Peer review of "Effects of a Digitally-Enabled Healthy Eating and Physical Activity Diabetes Prevention Peer Support Program on Weight over 6-Months"

_nutrients, 2025, doi:10.3390/nu17223599_

Round 1

Reviewer 1 Report (Previous Reviewer 3)

Comments and Suggestions for Authors

Thanks for the thorough revision; I consider the paper ready for acceptance.

Author Response

Thanks for the thorough revision; I consider the paper ready for acceptance.

Response:

Thank you for taking the time to review our changes.

Reviewer 2 Report (Previous Reviewer 2)

Comments and Suggestions for Authors

Review of the revised manuscript entitled “Effects of a digitally-enabled healthy eating and physical activity diabetes prevention peer support program on weight over 6-months” (Manuscript ID: nutrients-3991277)

This revised version has addressed many of the critical comments raised in the previous manuscript. The Authors have made great efforts to clarify the study design, refine the introduction, shorten the discussion, and clearly describe major limitations such as the small sample size, high dropout rate, and reliance on self-reported weight data. The manuscript is now clearer, more focused, and closer to the expected standard for nutrients. However, some important revisions are still needed before the study can be accepted.

Remaining critical comments

  • The manuscript continues to use inconsistent terminology (“pre-post study”, “feasibility study” and “program evaluation”? ). This should be harmonized in the title, abstract and methods.
  • The abstract and conclusions currently state that the program “achieved significant weight loss”. However, the uncontrolled design warrants rewording to avoid causal language. e.g. “…was associated with significant weight loss among participants who completed the program.”
  • The sample size (n=7 participants, n=1 facilitator) is still very small. It should be clearly indicated in both the abstract and the discussion that these qualitative results are exploratory and not generalizable.
  • The captions of the figures are unclear, especially in Figure 4 with statistical notation or description of error bars. These should be clarified!
  • The formatting of the standard p-value should also be checked (e.g., p < 0.001).
  • The Funding or Acknowledgements section should include a clear standard disclaimer, e.g., “The funders had no role in the design, collection, analysis, or interpretation of this study.”

The revised manuscript represents a clear improvement and adequately addresses most of the previous reviewer’s concerns. The remaining issues are primarily editorial and conceptual refinements related to study labeling and causal interpretation, rather than methodological errors.

Author Response

This revised version has addressed many of the critical comments raised in the previous manuscript. The Authors have made great efforts to clarify the study design, refine the introduction, shorten the discussion, and clearly describe major limitations such as the small sample size, high dropout rate, and reliance on self-reported weight data. The manuscript is now clearer, more focused, and closer to the expected standard for nutrients. However, some important revisions are still needed before the study can be accepted.

Remaining critical comments

  • The manuscript continues to use inconsistent terminology (“pre-post study”, “feasibility study” and “program evaluation”? ). This should be harmonized in the title, abstract and methods.
  • The abstract and conclusions currently state that the program “achieved significant weight loss”. However, the uncontrolled design warrants rewording to avoid causal language. e.g. “…was associated with significant weight loss among participants who completed the program.”
  • The sample size (n=7 participants, n=1 facilitator) is still very small. It should be clearly indicated in both the abstract and the discussion that these qualitative results are exploratory and not generalizable.
  • The captions of the figures are unclear, especially in Figure 4 with statistical notation or description of error bars. These should be clarified!
  • The formatting of the standard p-value should also be checked (e.g., p < 0.001).
  • The Funding or Acknowledgements section should include a clear standard disclaimer, e.g., “The funders had no role in the design, collection, analysis, or interpretation of this study.”

The revised manuscript represents a clear improvement and adequately addresses most of the previous reviewer’s concerns. The remaining issues are primarily editorial and conceptual refinements related to study labeling and causal interpretation, rather than methodological errors.

Response:

Thank you for your in-depth feedback. We have made the following changes to address your comments:

  1. In the abstract, introduction, and methods we have previously indicated that this is a pre-post study or have added this wording for clarity (lines 20, 98, 107-108). We have ensured that any mention of feasibility is related to the aims of the paper (in addition to assessing preliminary effects) and not the main study design.

Methods: A pre-post study of a digitally-enabled peer support initiative…”

“To address these gaps, this pre-post study aimed to assess the feasibility, and effect on weight…”

“This study employed a pre-post design…”

  1. Thank you for this suggestion. We have altered the wording in both the abstract and conclusion to not imply causation (lines 34 and 366-368):

“Preliminary results indicate that this digital program was associated with significant weight reduction…”

“In conclusion, this digitally-enabled program was feasible and acceptable for individuals at risk of diabetes in an inner-regional area of Sydney, with associated weight reductions providing preliminary indications that warrant further investigation.”

  1. We have added wording around the qualitative exit interviews to clarify that these findings are exploratory and not intended to be generalisable (lines 31-32 and 325-326):

“Exploratory qualitative analysis of exit interviews revealed challenges surrounding technology, website interaction, scheduling conflicts, data collection, and attrition.”

“We identified multiple challenges to the implementation and feasibility of this program through our exploratory qualitative analysis of exit interviews.”

  1. We have updated the figures (Figure 4 and S1) by adding a note to explain the contents of the figures more appropriately. We have also ensured that p values are formatted consistently throughout the manuscript.

  1. We have added the suggested text to confirm that the funders had no role in any aspect of the study other than providing financial support (lines 384-385):

“The funders had no role in the design, collection, analysis, or interpretation of this study.”

Reviewer 3 Report (Previous Reviewer 1)

Comments and Suggestions for Authors

The revised version has substantially improved. The introduction now provides a clearer rationale and appropriately highlights the gap concerning underserved inner-regional communities. The discussion has been expanded and includes a transparent acknowledgment of the main study limitations (pre–post design, high attrition, and reliance on self-reported data). The authors have also adopted a more cautious tone by referring to the results as preliminary, which is appropriate. However, as this is a feasibility study, conclusions should remain strictly limited to feasibility and acceptability rather than implying clinical efficacy. A brief reminder of this distinction in the conclusion would further strengthen the manuscript. Overall, the paper is well-written, relevant, and now suitable for publication after minor language polishing and final editing.

Author Response

The revised version has substantially improved. The introduction now provides a clearer rationale and appropriately highlights the gap concerning underserved inner-regional communities. The discussion has been expanded and includes a transparent acknowledgment of the main study limitations (pre–post design, high attrition, and reliance on self-reported data). The authors have also adopted a more cautious tone by referring to the results as preliminary, which is appropriate. However, as this is a feasibility study, conclusions should remain strictly limited to feasibility and acceptability rather than implying clinical efficacy. A brief reminder of this distinction in the conclusion would further strengthen the manuscript. Overall, the paper is well-written, relevant, and now suitable for publication after minor language polishing and final editing.

Response:

Thank you for your suggestions. We have made the following changes to address your comments:

  1. We have clarified the conclusion as to not imply clinical efficacy (lines 366-368). This now reads:

“In conclusion, this digitally-enabled program was feasible and acceptable for individuals at risk of diabetes in an inner-regional area of Sydney, with associated weight reductions providing preliminary indications that warrant further investigation.”

This manuscript is a resubmission of an earlier submission. The following is a list of the peer review reports and author responses from that submission.

Round 1

Reviewer 1 Report

Comments and Suggestions for Authors

Thank you for submitting your manuscript for review. The topic covered is highly relevant in the context of diabetes prevention, however, the current version presents some critical issues that compromise its suitability for publication. The study design, being pre-post without a control group, severely limits the ability to draw causal conclusions about the effectiveness of the intervention. Furthermore, the limited duration of six months does not allow for an assessment of the long-term sustainability of the effects, which is a central aspect of diabetes prevention strategies. The high dropout rate raises doubts about the actual generalizability and real impact of the program. Although the use of digital peer support represents an interesting opportunity, the novelty of the proposal appears limited, given that supported weight loss remains an already established concept. Difficulties related to engagement, technological problems, and inconsistent support from facilitators further complicate the interpretation of the results. A more cautious approach in the conclusions would be desirable, taking these limitations into account and presenting the results as preliminary. Finally, we recommend enriching the bibliographic context by highlighting the specific gaps that the study aims to fill or the original contribution it makes. I trust that these suggestions will be useful for improving the manuscript in the future and encourage the authors to strengthen the methodological structure, extend the duration of the study, and improve the critical discussion to enhance the importance of the work carried out.

Reviewer 2 Report

Comments and Suggestions for Authors

Review of the manuscript entitled „Effect of a digitally-enabled healthy eating and physical activity diabetes prevention peer support program on weight over 6-months” (Manuscript ID: nutrients-3915752)

This pre-post feasibility study evaluates a digitally supported lifestyle intervention in the inner city of Sydney, Australia, targeting individuals at increased risk of type 2 diabetes. The program includes educational videos, goal setting tools, performance logging, and monthly consultations with facilitators. Recruitment was largely through general practice. Of the 79 enrolled participants, 35 (43.8%) completed the program. Program completers achieved an average weight loss of 3.7 kg (p<0.001). Engagement with the platform, particularly tracking of outcomes, was correlated with greater weight loss among the individuals studied. Key challenges included dropout, technological barriers, limited facilitator confidence, and low engagement. The study concludes that such digital peer support programs are feasible and potentially effective, although retention and platform usability need refinement. The English is overall clear, grammatically correct, and professional.

Major Strengths

  • Relevant, timely topic addressing digital health and diabetes prevention in underserved populations.
  • Well-structured manuscript with adequate methodological detail.
  • Supplementary tables provide useful baseline comparisons and readiness-to-change analyses.
  • Demonstrates clinically meaningful weight reduction among completers.

Critical Comments

  • The continuity of the Introduction should be improved, avoiding too much detailed epidemiology, shortening the Introduction and focusing it on the rationale and objectives of the study!

  • The pre-post design without a control group severely limits the possibility of causal inference. This should be acknowledged more explicitly in both the Abstract and the Discussion.
  • Only 35 of the 79 participants were followed up, and attrition was skewed by gender, with more men completing the program. This attrition and potential selection bias should be emphasized in the Discussion.
  • The exit interviews (n=7 participants, n=1 moderator) are too limited to generalize. The authors should also acknowledge this limitation.
  • For some participants, the reliance on self-reported weights weakens validity. The proportion of self-reported and measured weights at follow-up should be clarified in the Methods and Results.
  • Consider shortening or reorganizing the long discussion section. To shorten the Discussion, it is useful to omit the repetitions detailing the general effects of digital interventions and peer support, as these have already been presented in the introduction, and a concise reference to the relevant literature is sufficient. The emphasis should be placed on the own results, i.e. the average weight loss of 3.7 kg and the fact that active platform use, especially “logging”, predicted better outcomes, while it is sufficient to briefly indicate the lower retention of female participants without separate analysis. The list of limitations can be combined and summarized in a few sentences, the constraints arising from the pre–post design, small sample, self-reported data and high dropout rates. It is not necessary to detail the technical problems either, it is enough to mention in general that participants and facilitators reported difficulties with usability, which may have reduced engagement. The final section should be brief, highlighting the main conclusions and then projecting the direction of future studies, such as larger samples, inclusion of a control group and measurement of additional clinical endpoints.

Questions

  • Would a hybrid model (digital + in-person peer support) improve retention, particularly for women who are less likely to complete their studies?
  • How could this platform be adapted for Australia’s culturally diverse or Indigenous population, who are often at higher risk for diabetes but are underrepresented here?
  • In addition to weight, what cardiometabolic outcomes (HbA1c, blood pressure, lipid profile) should be measured in the future to demonstrate broader clinical benefits?

I would find it useful to add answers to these questions to the Discussion section!

The manuscript presents useful and clinically relevant findings. However, the authors should refine limitations, clarify methods, and slightly restructure the text to strengthen transparency and readability. 

Reviewer 3 Report

Comments and Suggestions for Authors

The authors present a single-arm intervention study assessing the effectiveness and acceptability of a lifestyle support program.

The overall rationale is clear.

Introduction: Fine.

Methods: 

Study is compromised by low adherence, high drop-out rate, but this is the result targetting by feasibility assessment. Ergo, no argument against publication.

Results: The flow chart needs improvement; excluded patients / drop-outs (etc.) should always split off to the right; remaining participants follow the flow downwards.

Primary outcome of weight loss could additionally be assessed by ITT principles in order to clarify, whether drop-outs and non-adherence relevantly affect efficacy of the novel treatment option.

Discussion: Fine.